# HELLINGER DISTANCE CONSTRAINED REGRESSION

## ABSTRACT

This paper introduces an off-policy reinforcement learning method that uses Hellinger distance between sampling policy (from what samples were collected) and current policy (policy being optimized) as a constraint. Hellinger distance squared multiplied by two is greater than or equal to total variation distance squared and less than or equal to Kullback-Leibler divergence, therefore a lower bound for expected discounted return for the new policy is improved compared to the lower bound for training with KL. Also, Hellinger distance is less than or equal to 1, so there is a policy-independent lower bound for expected discounted return. HDCR is capable of training with Experience Replay, a common setting for distributed RL when collecting trajectories using different policies and learning from this data centralized. HDCR shows results comparable to or better than Advantage-weighted Behavior Model and Advantage-Weighted Regression on MuJoCo tasks using tiny offline datasets collected by random agents. On bigger datasets (100k timesteps) obtained by pretrained behavioral policy, HDCR outperforms ABM and AWR methods on 3 out of 4 tasks.

## 1 INTRODUCTION

Policy gradient algorithms are methods of model-free reinforcement learning that optimize policy through differentiating expected discounted return. Despite the simplicity, to converge, these methods should stay on-policy because of the first-order approximation of state visitation frequencies. This issue makes agents learn through trial-and-error, using data only once.

To make policy gradient updates more off-policy, we can add a constraint on the update to decrease the step size if the current policy is too far from the sampling policy. One of the first methods was to add total variation distance squared as a constraint for mixture policies. Later it was proven that there is a lower bound for new policy's expected discounted return (Kakade & Langford, 2002). Recently it was proven that this lower bound exists for all types of updates (Schulman et al., 2015).

Next, total variation distance squared was replaced by Kullback-Leibler divergence that is greater than or equal to the previous one (Pinker's inequality (Levin & Peres, 2017)), so that the lower bound was decreased (Schulman et al., 2015). Using Lagrangian, have been derived off-policy method called Advantage-Weighted Regression (Peng et al., 2019), which also used KL as a constraint.

This article proposes a new method whose lower bound of expected discounted return is greater than or equal to the bound with KL. We achieve this by replacing total variation distance by Hellinger distance, which decreases lower bound. Therefore strictness stays the same. Then we derive an off-policy method called Hellinger Distance Constrained Regression using the new constraint. It can be used on discrete and continuous action spaces since derivation uses Lebesgue integrals rather than a summation or Riemann integrals.

## 2 PRELIMINARIES

To better present the problem, we start from basic definitions, go through the history of improvements, and then describe the disadvantages of using KL divergence as a constraint.

We consider an infinite-horizon discounted Markov decision process (MDP), defined by the tuple $(\mathcal{S}, \mathcal{A}, P, r, \rho_0, \gamma)$, where $\mathcal{S}$ is a set of states (finite or infinite), $\mathcal{A}$ is a set of actions (finite or infinite), $P : \mathcal{S} \times \mathcal{A} \times \mathcal{S} \to \mathbb{R}$ is the transition probability distribution, $r : \mathcal{S} \to \mathbb{R}$ is

the reward function, $\rho_0 : \mathcal{S} \to \mathbb{R}$ is the distribution of the initial state $s_0$, and $\gamma \in (0, 1)$ is the discount factor.

Let $\pi$ denote a stochastic policy $\pi : \mathcal{S} \times \mathcal{A} \to [0, 1]$, and then its expected discounted return is:

$$\eta(\pi) = \mathbb{E}_{s_0, a_0, \dots} \left[ \sum_{t=0}^{\infty} \gamma^t r(s_t) \right], \text{ where}$$

$$s_0 \sim \rho_0(\cdot), \ a_t \sim \pi(\cdot|s_t), \ s_{t+1} \sim P(\cdot|s_t, a_t). \tag{1}$$

This paper uses state-action value function $Q_\pi$, state value function $V_\pi$, and advantage function $A_\pi$ with the following definitions:

$$Q_\pi(s_t, a_t) = \mathbb{E}_{s_{t+1}, a_{t+1}, \dots} \left[ \sum_{l=0}^{\infty} \gamma^l r(s_{t+l}) \right]$$

$$V_\pi(s_t) = \mathbb{E}_{a_t, s_{t+1}, a_{t+1}, \dots} \left[ \sum_{l=0}^{\infty} \gamma^l r(s_{t+l}) \right] \tag{2}$$

$$A_\pi(s_t, a_t) = Q_\pi(s_t, a_t) - V_\pi(s_t).$$

Let $\rho_\pi(s)$ be unnormalized visitation frequencies of state $s$ where actions are chosen according to $\pi$:

$$\rho_\pi(s) = \sum_{t=0}^{\infty} \gamma^t P(s_t = s). \tag{3}$$

Following identity expresses the expected return of another policy $\tilde{\pi}$ in terms of the advantage over $\pi$, accumulated over states (see Schulman et al. (2015) for proof):

$$\eta(\tilde{\pi}) = \eta(\pi) + \int \rho_{\tilde{\pi}}(s) \int \tilde{\pi}(a|s) A_\pi(s, a) \, da \, ds. \tag{4}$$

In approximately optimal learning, we replace state visitation frequency $\rho_{\tilde{\pi}}$ by $\rho_\pi$, since this drastically decrease optimization complexity:

$$L_\pi(\tilde{\pi}) = \eta(\pi) + \int \rho_\pi(s) \int \tilde{\pi}(a|s) A_\pi(s, a) \, da \, ds. \tag{5}$$

Let $\pi_{old}$ denote current policy, then the lower bound for the expected discounted return for the new policy $\pi_{new}$ will be (see Schulman et al. (2015) for proof):

$$\eta(\pi_{new}) \geq L_{\pi_{old}}(\pi_{new}) - \frac{4\epsilon\gamma}{(1-\gamma)^2} \alpha^2$$

$$\text{where } \epsilon = \max_{s,a} |A_\pi(s, a)|,$$

$$\alpha = \max_s D_{TV}(\pi_{old}(\cdot|s)||\pi_{new}(\cdot|s)), \tag{6}$$

$$D_{TV}(\pi_{old}(\cdot|s)||\pi_{new}(\cdot|s)) = \frac{1}{2} \int |\pi_{old}(a|s) - \pi_{new}(a|s)| \, da.$$

Theoretical Trust-Region Policy Optimization algorithm relays on Pinsker's inequality (see (Tsybakov, 2009) for proof):

$$D_{KL}(\pi_{old}(\cdot|s)||\pi_{new}(\cdot|s)) \geq D_{TV}(\pi_{old}(\cdot|s)||\pi_{new}(\cdot|s))^2$$

$$\text{where } D_{KL}(\pi_{old}(\cdot|s)||\pi_{new}(\cdot|s)) = \int \pi_{old}(a|s) \log \frac{\pi_{old}(a|s)}{\pi_{new}(a|s)} \, da. \tag{7}$$

To retain strictness and decrease calculation complexity, total variation distance squared was replaced with Kullback-Leibler divergence $D_{KL}(\pi_{old}(\cdot|s)||\pi_{new}(\cdot|s))$:

$$\eta(\pi_{new}) \geq L_{\pi_{old}}(\pi_{new}) - C D_{KL}^{max}(\pi_{old}||\pi_{new})$$
$$\text{where } \epsilon = \max_{s,a} |A_\pi(s,a)|,$$
$$C = \frac{4\epsilon\gamma}{(1-\gamma)^2}, \tag{8}$$
$$D_{KL}^{max}(\pi_{old}||\pi_{new}) = \max_s D_{KL}(\pi_{old}(\cdot|s)||\pi_{new}(\cdot|s)).$$

However, this replacement greatly decreases the lower bound for the expected discounted return for the new policy. Moreover, Kullback-Leibler divergence has no upper bound. Therefore we have no policy-independent lower bound for this type of update.

## 3 HELLINGER DISTANCE IN POLICY OPTIMIZATION

We can improve lower bound (compared to KL) by replacing $D_{TV}(\pi_{old}(\cdot \mid s) \mid\mid \pi_{new}(\cdot \mid s))$ with Hellinger distance $H(\pi_{old}(\cdot|s) \mid\mid \pi_{new}(\cdot \mid s))$:

$$H(\pi_{old}(\cdot|s) \mid\mid \pi_{new}(\cdot|s))^2 = 1 - \int \sqrt{\pi_{old}(a|s)\,\pi_{new}(a|s)}\,da \tag{9}$$

Theorem 1 (see Appendix A or (Tsybakov, 2009, section 2.4) for proof) proves correctness and improvement (compared to KL) to the lower bound.

Let $p(v)$ and $q(v)$ be two probability density functions then:

$$D_{TV}(p(\cdot) \mid\mid q(\cdot))^2 \leq 2H(p(\cdot) \mid\mid q(\cdot))^2 \leq D_{KL}(p(\cdot) \mid\mid q(\cdot)) \tag{10}$$

Replacing $p(v)$ and $q(v)$ with $\pi_{old}(\cdot \mid s)$ and $\pi_{new}(\cdot \mid s)$ respectively, new lower bound follows:

$$\eta(\pi_{new}) \geq L_{\pi_{old}}(\pi_{new}) - \frac{8\epsilon\gamma}{(1-\gamma)^2}\alpha^2$$
$$\text{where } \epsilon = \max_{s,a} |A_\pi(s,a)|, \tag{11}$$
$$\alpha = \max_s H(\pi_{old}(\cdot|s)||\pi_{new}(\cdot|s))$$

It is worth to note that $H(\pi_{old}(\cdot \mid s) \mid\mid \pi_{new}(\cdot \mid s)) \leq 1$.

## 4 HELLINGER DISTANCE CONSTRAINED REGRESSION (HDCR)

We could use presented lower bound as in TRPO, but instead, we derive an offline regression algorithm by introducing the following optimization problem where $\mu$ is the sampling policy:

$$\arg\max_\pi \int \rho_\mu(s) \int \pi(a|s) A_\mu(s,a)\,da\,ds$$
$$\text{s.t. } \int \rho_\mu(s) H(\pi(\cdot|s)||\mu(\cdot|s))\,ds \leq \epsilon, \tag{12}$$
$$\int \pi(a|s)\,da = 1, \ \forall s \in \mathcal{S}.$$

Constructing Lagrangian, differentiating it with respect to $\pi$, and solving for $\pi$ gives us the following optimal policy (see Appendix B for derivation):

$$\pi^*(a|s) = \mu(a|s)\frac{\beta^2}{(\beta - 2A_\mu(s,a))^2}, \text{ where } \beta \text{ is a Lagrangian multiplier.} \tag{13}$$

Constructing a regression problem of KL divergence between optimal policy $\pi^*$ and current policy $\pi$ and simplifying gives us the following supervised regression problem (see Appendix B for derivation):

$$\arg\max_\pi \mathbb{E}_{s\sim\rho_\mu(\cdot)} \mathbb{E}_{a\sim\mu(\cdot|s)} \log \pi(a|s)\frac{1}{(\beta - A_\mu(s,a))^2} \tag{14}$$

Using notation of ABM paper (Siegel et al., 2020) "advantage-weighting" function is $f(A(s,a)) = \frac{1}{(\beta - A(s,a))^2}$.

If we use HDCR with Experience Replay, in equation 14, we replace $\mu(\cdot|s)$ in expectation $\mathbb{E}_{a \sim \mu(\cdot | s)}$ and advantage function $A_\mu(s, a)$. Let $\Pi = \{\pi_i, \pi_{i+1}, ..., \pi_{i+N}\}$ be a set of sampling policies from which actions were sampled, $w(\pi_i)$ probability of selecting policy $\pi_i$, then:

$$\mu(s,a) = \int_\Pi w(\pi)\rho_\pi(s)\pi(a|s)\, d\pi$$

$$A_\mu(s,a) = \frac{\int_\Pi w(\pi)\rho_\pi(s)\left(Q_\pi(s,a) - V_\pi(s)\right)\, d\pi}{\int_\Pi w(\pi)\rho_\pi(s)\, d\pi} \tag{15}$$

$$\overline{V}(s) = \frac{\int_\Pi w(\pi)\rho_\pi(s)V_\pi(s)\, d\pi}{\int_\Pi w(\pi)\rho_\pi(s)\, d\pi}$$

The proof will repeat the proof for AWR with Experience Replay (Peng et al., 2019).

Practically we simply sample uniformly from the replay buffer and using a value function estimator. This type of sampling provides us an approximation of expectation and state value function.

Let $\mathcal{D}$ denote a set of stored trajectories (replay buffer), $A(s, a; \phi)$ is an advantage function parameterized by vector $\phi$.

The most popular method to obtain $A(s, a; \phi)$ for offline learning is to use state-action function estimator $Q(s, a; \phi)$ parameterized by $\phi$:

$$A(s,a;\phi) = Q(s,a;\phi) - \int \pi(a'|s;\theta_k)Q(s,a';\phi_k)da' \tag{16}$$

Despite $Q(s, a; \phi)$ being closer to the expectation of discounted return following policy $\pi$ (because of "taking" the first action according to $\pi$ rather then $\mu$), we found Monte-Carlo return more efficient on tiny offline datasets. Greater performance using MC return can be explained by a lack of experience "produced" by certain actions.

Monte-Carlo estimation of $A(s, a; \phi)$ can be described as follows where $\mathcal{R}^{\mathcal{D}}_{s_t,a_t} = \sum_{l=0}^{T} \gamma^l r_{t+l}$ and $V(s; \phi)$ is a state value function estimator parameterized by vector $\phi$:

$$A(s,a;\phi) = \mathcal{R}^{\mathcal{D}}_{s,a} - V(s;\phi) \tag{17}$$

Also, we can use Generalized Advantage Estimation (Schulman et al., 2016), where $\gamma \in [0, 1]$ and $\delta_t^{\phi_k}$ is a one-step temporal difference for state $s_t$ calculated using old vector $\phi_k$:

$$\delta_t^{\phi_k} = r_t + \gamma V(s_{t+1};\phi_k) - V(s_t;\phi_k)$$

$$\hat{A}(s_t,a_t;\phi_k) = \sum_{l=0}^{T}(\gamma\lambda)^l \delta_{t+l}^{\phi_k} \tag{18}$$

$$A(s_t,a_t;\phi) = \hat{A}(s_t,a_t;\phi_k) + V(s_t;\phi_k) - V(s_t;\phi)$$

Finally, we propose the following reinforcement learning algorithm:

---
**Algorithm 1:** Hellinger Distance Constrained Regression

---
$\theta_1 \leftarrow$ random initial weights
$\mathcal{D} \leftarrow \emptyset$
**for** *iteration* $k = 1, ..., k_{max}$ **do**
    add trajectories $\{\tau_i\}$ sampled via $\pi_{\theta_k}$ to $\mathcal{D}$
    $\phi_{k+1} \leftarrow \arg\min_\phi \mathbb{E}_{s,a\sim\mathcal{D}} A^2(s,a;\phi)$
    $\theta_{k+1} \leftarrow \arg\max_\theta \mathbb{E}_{s,a\sim\mathcal{D}} \left[\log \pi_\theta(a|s)\frac{1}{(\beta-A(s,a;\phi_k))^2}\right]$
**end**

---

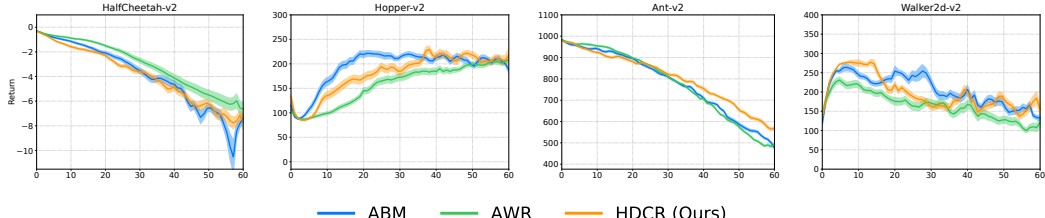

Figure 1: Learning curves of ABM, AWR, and HDCR averaged across results of learning from 10 different datasets of 10k timestamps (also 5 seeds used to generate each dataset).

| Task | ABM | AWR | HDCR (Ours) |
|------|-----|-----|-------------|
| Ant-v2 | $468 \pm 72$ | $495 \pm 84$ | $\mathbf{569 \pm 64}$ |
| HalfCheetah-v2 | $\mathbf{-7 \pm 3}$ | $\mathbf{-7 \pm 4}$ | $-8 \pm 6$ |
| Hopper-v2 | $184 \pm 72$ | $209 \pm 31$ | $\mathbf{223 \pm 141}$ |
| Walker2d-v2 | $132 \pm 76$ | $125 \pm 92$ | $\mathbf{145 \pm 117}$ |

Table 1: Final returns for different algorithms, with $\pm$ corresponding to one standard deviation of the average return across 10 datasets of 10k timestamps.

## 5 EXPERIMENTS

In our experiments, we evaluate the algorithm on MuJoCo (Todorov et al., 2012) tasks.

### 5.1 TINY DATASETS

For evaluating on extremely small datasets we use setting inspired by Behavioral Modelling Priors for Offline Reinforcement Learning paper (Siegel et al., 2020) but instead of using actions from a behavioral policy we use random actions while generating buffer.

First, we collect 2048 timestamps or more, until episode termination (whichever occurred later), from each of 5 seeds using random actions. Then we load collected trajectories to a replay buffer for the agent training. Separate networks with the same architecture (except the last layer) represent the policy and value function and consist of 2 hidden layers of 256 ELU units. Each train iteration uses only old data obtained by random agents. Each iteration, the value function is updating with 5 gradient steps and policy with 50 steps using a uniformly sampled batches of 512 samples using all data from the replay buffer. Learning rates for Adam optimizer are $2 \times 10^{-3}$ and $2 \times 10^{-4}$ for critic and actor, respectively.

We compare 3 different "advantage-weight" functions $f(A(s,a))$:

- Hellinger Distance Constrained Regression, where $f(A(s,a)) = \frac{1}{(\beta - A(s,a))^2}$;
- Advantage-weighted Behavior Model, where $f(A(s,a)) = I_{A(s,a)>0}$, $I_{x>0} = 1$ if $x > 0$ otherwise $I_{x>0} = 0$;
- Advantage-Weighted Regression, where $f(A(s,a)) = \exp(\frac{1}{\beta} A(s,a))$.

AWR method uses $\beta = 1.0$ as it is in implementation released by authors. HDCR uses $\beta = 1.0$. For TD($\lambda$) we use $\lambda = 0.95$.

On simple tasks as Hopper-v2, all methods are able to learn (Figure 1), and HDCR shows slightly better results (Table 1). While on difficult tasks as Ant-v2, all algorithms do not improve their results through iterations. Moreover, evaluation returns decrease.

### 5.2 LARGE DATASETS

Next, we perform tests using buffers with the size of 100k timesteps. Buffer is filled by running a pretrained behavioral policy. This setting replicates the setting from Off-Policy Deep Reinforce-

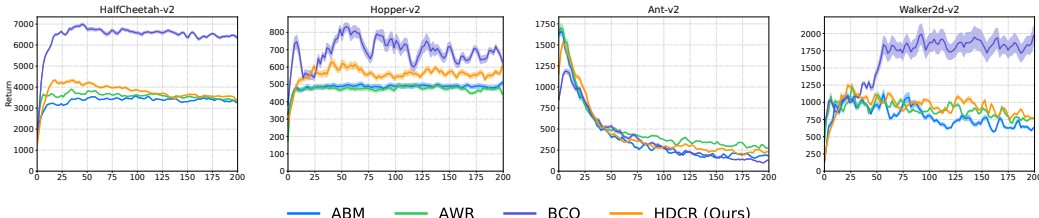

Figure 2: Curves of BCQ, ABM, AWR, and HDCR evaluation results averaged across 3 seeds.

| Task | ABM | AWR | HDCR (Ours) |
|------|-----|-----|-------------|
| Ant-v2 | $179 \pm 45$ | $\mathbf{281 \pm 25}$ | $217 \pm 54$ |
| HalfCheetah-v2 | $3358 \pm 76$ | $3365 \pm 193$ | $\mathbf{3485 \pm 205}$ |
| Hopper-v2 | $495 \pm 115$ | $478 \pm 94$ | $\mathbf{576 \pm 123}$ |
| Walker2d-v2 | $592 \pm 139$ | $762 \pm 64$ | $\mathbf{805 \pm 62}$ |

Table 2: Final returns for different algorithms, with $\pm$ corresponding to one standard deviation of the average return across 3 seeds.

ment Learning without Exploration (Fujimoto et al., 2019) paper. Therefore we also provide results of BCQ method achieved by authors' implementation trained from the same datasets. For calculating advantage we use equation 16 where we approximate integral by taking mean of 10 Q-values obtained by 10 actions sampled from the policy.

While BCQ outperforms all the presented methods, it uses a gradient of Q-value function in actor training, which provides better generalization. This provides better results on evaluation but affects stability and performance. Against other methods that update the policy function directly, HDCR shows better results on 3 environments out of 4 (Figure 2 and Table 2).

## 6 DISCUSSION

We theoretically proved that Hellinger distance improves the lower bound of expected discounted return compared to Kullback-Leibler divergence and proposed a simple off-policy reinforcement learning method that uses Hellinger distance as a constraint. The expected discounted return for a new policy now has a policy-independent lower bound. This bound guarantees that return will not decrease in "one" shot.

Experiments show that HDCR outperforms both ABM and AWR on tiny datasets obtained by random agents. This performance proves the efficiency of using Hellinger distance by allowing bigger step sizes, retaining lower bound.

On bigger datasets, HDCR shows comparable or better results than AWR and ABM.

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

## A    THEOREM 1 PROOF

Let $(\Omega, \mathcal{A})$ be a measurable space, $P$ and $Q$ be two probability measures on that space, $v$ be a $\sigma$-finite measure on $(\Omega, \mathcal{A})$ such that $P \ll v$ ($P(A) = 0$ for any $A \in \mathcal{A}$ such that $\mu(A) = 0$) and $Q \ll v$. And let us denote Radon–Nikodym derivatives (in our derivations we can assume them as probability density functions) as follows:

$$p = \frac{dP}{dv}$$
$$q = \frac{dQ}{dv} \tag{19}$$

So, $p$ and $q$ satisfy following conditions:

$$P(A) = \int_A p(v) \, dv \ \ \forall A \in \mathcal{A}$$
$$Q(A) = \int_A q(v) \, dv \ \ \forall A \in \mathcal{A} \tag{20}$$

Then we define distances between probability measures:

$$D_{TV}(P \, || \, Q) = \sup_{A \in \mathcal{A}} |P(A) - Q(A)|$$
$$H(P \, || \, Q)^2 = \frac{1}{2} \int (\sqrt{p(v)} - \sqrt{q(v)})^2 \, dv = 1 - \int \sqrt{p(v)q(v)} \, dv \tag{21}$$
$$D_{KL}(P \, || \, Q) = \int \log \frac{dP}{dQ} \, dP = \int p(v) \log \frac{p(v)}{q(v)} \, dv$$

Following Lemmas will be used in Theorem 1 proof:

**Lemma 1.** *Given two probability distributions $p$ and $q$ total variance divergence can be calculated as follows:*

$$D_{TV}(P \, || \, Q) = \sup_{A \in \mathcal{A}} |P(A) - Q(A)| = \frac{1}{2} \int |p(v) - q(v)| \, dv \tag{22}$$

*Proof.* Let $B = \{v : P(v) > Q(v)\}$, $B^c = \{v : P(v) \le Q(v)\}$ and $A \in \mathcal{A}$:
$$P(A) - Q(A) \le P(A \cap B) - Q(A \cap B) \le P(B) - Q(B)$$
$$P(A) - Q(A) \le Q(A \cap B^c) - P(A \cap B^c) \le Q(B^c) - P(B^c)$$
$$\sup_{A \in \mathcal{A}} |P(A) - Q(A)| = P(B) - Q(B) = Q(B^c) - P(B^c) \tag{23}$$
$$D_{TV}(P \, || \, Q) = \frac{1}{2} [P(B) - Q(B) + Q(B^c) - P(B^c)]$$
$$= \frac{1}{2} \int |p(v) - q(v)| \, dv$$

$\square$

**Lemma 2.**

$$D_{TV}(P \, || \, Q) = 1 - \int \min(p(v), q(v)) \, dv \tag{24}$$

*Proof.*

$$D_{TV}(P \, || \, Q) = \frac{1}{2} \int |p(v) - q(v)| \, dv = \int_{\{v: \, p(v) > q(v)\}} [p(v) - q(v)] \, dv$$
$$= 1 - \int_{\{v: \, p(v) < q(v)\}} p(v) \, dv - \int_{\{v: \, p(v) > q(v)\}} q(v) \, dv \tag{25}$$
$$= 1 - \int \min(p(v), q(v)) \, dv$$

$$\square$$

**Lemma 3.**
$$\int \max(p(v), q(v)) \, dv + \int \min(p(v), q(v)) \, dv = 2 \tag{26}$$

*Proof.* Rewriting left part in 4 integrals over following sets $\{\max(p(v), q(v)) = p(v)\}$, $\{\max(p(v), q(v)) = q(v)\}$, $\{\min(p(v), q(v)) = q(v)\}$ and $\{\min(p(v), q(v)) = q(v)\}$ and stacking integrals back gives us $P(\Omega) + Q(\Omega) = 2$. $\square$

**Theorem 1.** *For any two probability density functions $p$ and $q$, the following double inequality is true:*
$$D_{TV}(p \,||\, q)^2 \leq 2H(p \,||\, q)^2 \leq D_{KL}(p \,||\, q) \tag{27}$$

*Proof.* First inequality can be proved as follows:

$$
\begin{aligned}
(1 - H(P \,||\, Q)^2)^2 &= \left[ \int \sqrt{p(v)q(v)} \, dv \right]^2 \\
&= \left[ \int \sqrt{\min(p(v), q(v)) \max(p(v), q(v))} \, dv \right]^2 \\
&\leq \int \min(p(v), q(v)) \, dv \int \max(p(v), q(v)) \, dv \\
&= \int \min(p(v), q(v)) \, dv \left[ 2 - \int \min(p(v), q(v)) \, dv \right] \\
&= (1 - D_{TV}(P \,||\, Q))(1 + D_{TV}(P \,||\, Q)) \\
&= 1 - D_{TV}(P \,||\, Q)^2 \\
D_{TV}(P \,||\, Q)^2 &\leq H(P \,||\, Q)^2 (2 - H(P \,||\, Q)^2) \\
D_{TV}(P \,||\, Q)^2 &\leq 2H(P \,||\, Q)^2
\end{aligned}
\tag{28}
$$

Proof of the second inequality:

$$
\begin{aligned}
D_{KL}(P \,||\, Q) &= \int p(v) \log \frac{p(v)}{q(v)} \, dv = 2 \int p(v) \log \sqrt{\frac{p(v)}{q(v)}} \, dv \\
&= -2 \int p(v) \log \left( \left[ \sqrt{\frac{q(v)}{p(v)}} - 1 \right] + 1 \right) dv \\
&\geq -2 \int p(v) \left[ \sqrt{\frac{q(v)}{p(v)}} - 1 \right] dv = -2 \int \left[ \sqrt{q(v)p(v)} - 1 \right] dv \\
&= 2 \left[ 1 - \int \sqrt{p(v)q(v)} \, dv \right] = 2H(P \,||\, Q)^2 \\
D_{KL}(P \,||\, Q) &\geq 2H(P \,||\, Q)^2
\end{aligned}
\tag{29}
$$

$$\square$$

## B   HDCR DERIVATION

Given optimization problem:

$$
\begin{aligned}
\arg\max_{\pi} &\int \rho_\mu(s) \int \pi(a|s) A_\mu(s, a) \, da \, ds \\
\text{s.t.} &\int \rho_\mu(s) H(\pi(\cdot|s) || \mu(\cdot|s)) \, ds \leq \epsilon, \\
&\int \pi(a|s) \, da = 1, \ \forall s \in \mathcal{S}.
\end{aligned}
\tag{30}
$$

Constructing Lagrangian where $\alpha : \mathcal{S} \to \mathbb{R}$ is a function for obtaining Lagrange multiplier for every state, $\beta$ is also a Lagrange multiplier:

$$
\begin{aligned}
\mathcal{L}(\pi, \beta, \alpha) = & \int_s \rho_\mu(s) \int_a \pi(a|s) A_\mu(s, a) \, da \, ds \\
& + \beta \left( \epsilon - \int_s \rho_\mu(s) \left[ 1 - \int \sqrt{\pi(a|s)\mu(a|s)} \, da \right] ds \right) \\
& + \int_s \alpha_s \left( 1 - \int_a \pi(a|s) \, da \right) ds
\end{aligned}
\tag{31}
$$

Differentiating with respect to $\pi(a|s)$ gives us following result:

$$
\frac{\partial \mathcal{L}}{\partial \pi(a|s)} = \rho_\mu(s) A_\mu(s, a) + \beta \rho_\mu(s) \frac{\mu(a|s)}{2\sqrt{\pi(a|s)\mu(a|s)}} - \alpha_s = 0
\tag{32}
$$

Solving for $\pi(a|s)$:

$$
\begin{aligned}
\beta \rho_\mu(s) \frac{\mu(a|s)}{2\sqrt{\pi(a|s)\mu(a|s)}} &= \alpha_s - \rho_\mu(s) A_\mu(s, a) \\
\frac{\sqrt{\mu(a|s)}}{\sqrt{\pi(a|s)}} &= 2 \frac{\frac{\alpha_s}{\rho_\mu(s)} - A_\mu(s, a)}{\beta}
\end{aligned}
\tag{33}
$$

Substituting $\pi(a|s) = \mu(\cdot|s)$ and taking expectation over actions taken according to $\mu$ gives us expression for $\alpha_s$:

$$
1 = 2 \frac{\frac{\alpha_s}{\rho_\mu(s)} - A_\mu(s, a)}{\beta}
$$

$$
\mathbb{E}_{a \sim \mu(\cdot|s)} \alpha_s = \alpha_s = \rho_\mu(s) \mathbb{E}_{a \sim \mu(\cdot|s)} \left[ \frac{1}{2}\beta + A_\mu(s, a) \right] = \frac{1}{2}\beta \rho_\mu(s)
\tag{34}
$$

Then optimal policy $\pi^*(a|s)$ can be written as follows:

$$
\pi^*(a|s) = \mu(a|s) \frac{\beta^2}{(\beta - 2A_\mu(s, a))^2}
\tag{35}
$$

To obtain regression problem we construct Kullback–Leibler divergence between optimal policy $\pi^*$ and current policy $\pi$:

$$
\begin{aligned}
& \arg\min_\pi \mathbb{E}_{s \sim \rho_\mu(\cdot)} D_{KL}(\pi^*(\cdot|s) \,||\, \pi(\cdot|s)) \\
&= \arg\min_\pi \mathbb{E}_{s \sim \rho_\mu(\cdot)} \int \pi^*(a|s) \log \frac{\pi^*(a|s)}{\pi(a|s)} \, da \\
&= \arg\min_\pi \mathbb{E}_{s \sim \rho_\mu(\cdot)} \int \mu(a|s) \frac{\beta^2}{(\beta - 2A_\mu(s, a))^2} \log \frac{\mu(a|s) \frac{\beta^2}{(\beta - 2A_\mu(s,a))^2}}{\pi(a|s)} \, da \\
&= \arg\min_\pi \mathbb{E}_{s \sim \rho_\mu(\cdot)} \mathbb{E}_{a \sim \mu(\cdot|s)} \frac{1}{(\beta - 2A_\mu(s, a))^2} \left[ \log \left( \mu(a|s) \frac{\beta^2}{(\beta - 2A_\mu(s, a))^2} \right) - \log \pi(a|s) \right] \\
&= \arg\max_\pi \mathbb{E}_{s \sim \rho_\mu(\cdot)} \mathbb{E}_{a \sim \mu(\cdot|s)} \log \pi(a|s) \frac{1}{(\beta - 2A_\mu(s, a))^2} \\
&= \arg\max_\pi \mathbb{E}_{s \sim \rho_\mu(\cdot)} \mathbb{E}_{a \sim \mu(\cdot|s)} \log \pi(a|s) \frac{1}{(\beta - A_\mu(s, a))^2}
\end{aligned}
\tag{36}
$$

Regression problem follows:

$$
\arg\max_\pi \mathbb{E}_{s \sim \rho_\mu(\cdot)} \mathbb{E}_{a \sim \mu(\cdot|s)} \log \pi(a|s) \frac{1}{(\beta - A_\mu(s, a))^2}
\tag{37}
$$

