# OpenReview forum: "Hellinger Distance Constrained Regression"
_ICLR.cc/2021/Conference — Reject_

### Official Review · AnonReviewer3 · 2020-10-27
**Hellinger Distance Constrained Regression**

**Rating:** 4
**Confidence:** 5

**Review:**

Summary:

This paper proposes a supervised learning for off-policy reinforcement learning. It exploits the Hellinger distance instead of KL divergence. Thus it achieves tighter lower bound of the expected culmulative return than that using the KL divergence. Moreover, the new lower bound is policy independent. The experimental results show that the proposed method slightly outperforms other baselines when only small amount of data are given, while the algorithms fail to learn on several environments.


Reasons for score:

Though it has some advantages, I vote to reject this paper. This is because, it has low novelty, the experiments are wrongly designed, and thus it is hard to believe the results. The specific details are below.


Pros

+ Hellinger divergence is used instead of KL divergence, and thus the lower bound become tighter than that using KL divergence.

+ The loss function for policy can be derived by theory


Cons

- Changing KL distance to Hellinger divergence has low novelty.  Also, the derivation of the loss function using Hellinger distance isn't difficult.  Hellinger distance and KL divergence are all under the class of Amari alpha-divergence. When alpha = +/- 1, Amari alpha-divergence becomes KL and when alpha=0, Amari alpha-divergence becomes the Hellinger distance = integral [sqrt(p) - sqrt(q)]^2 dx.   Indeed, HD is symmetric and satisfies the axioms of distance. Basically, when we consider the HD on the space of probability distribution, we consider Euclidean geometry on the space of probability distribution, whereas the KLD induces the Boltzman interpretation, i.e., p ~ exp( -KLD).

- In addition to the issue of significance in novelty,  the numerical results show that the performance improvement is insignificant or negligible. .

- The experiments used data sampled by random policies or first few samples of on-policy data, but I think that this is a little strange training setting. Most of the previous works in this line use samples at a certain performance (NOT DRAWN BY RANDOM POLICY). For example, in ABM paper[1], it used first 10,000 episodes (if the length of an episode is 1,000, it uses first 10 million samples), or first 2,000 episodes (first 2 million samples) to show its performance when it uses high performed samples, or low performed samples, respectively. These contain good performed samples relative to the random samples. However, experiments in this paper use almost random samples to train policies. We cannot expect a good policy at a certain performance using these random samples. This expectation is also shown in the results. Some learning curves go down as learning proceeds, and this  means that the learning fails on these environments. If the proposed method learns successfully while the others fail to learn, it is a meaningful result, but it is not, otherwise. I think that the authors should evaluate performance using better samples to prove that the proposed method outperforms others.


Reference

[1] Noah Siegel, et al. Keep doing what worked: Behavior modelling priors for offline reinforcement learning. In International Conference on Learning Representations, 2020.

---

> ### Comment · ~Wu_Zheng3 · 2022-06-12
> **R1**
>
> TEST

---

> ### Comment · ~Wu_Zheng3 · 2022-06-12
> **R2**
>
> TEST AGAIGN

---

> ### Comment · ~Wu_Zheng3 · 2022-06-12
> **R3**
>
> TEST

---

### Official Review · AnonReviewer1 · 2020-10-27
**Insufficient contribution and experimental validation**

**Rating:** 3
**Confidence:** 4

**Review:**

The authors propose the use of the Hellinger distance instead of KL divergence to constrain new policies to remain close to the behavior policy. The technical aspects are straightforward, noting that Hellinger provides tighter bounds on total variation than KL, and can straightforwardly be plugged into the CPI/TRPO bounds for policy improvement. They also propose an offline reinforcement learning algorithm based on enforcing a Hellinger constraint to the data policy, deriving iterative optimization procedure, and evaluate it on offline

I find the experimental evaluation highly lacking. It seems with the datasets and envs evaluated, policy performance actually *drops* as policy optimization is conducted, so it is not clear to me that these evaluations actually provide meaningful information towards which methods perform better in scenarious where we would want to use offline RL. I would like to see much more extensive evaluation of this method compared to other offline RL algorithms like BCQ https://arxiv.org/abs/1812.02900, BRAC https://arxiv.org/abs/1911.11361, or CQL https://arxiv.org/abs/2006.04779, over a much wider variety of datasets.

In general, I'm not convinced that simply using the Hellinger distance instead of KL will lead to significant improvements on its own, given that in the BRAC paper, the authors experimented with different trust regions including Wasserstein, MMD, and KL and didn't find huge differences in the tested domains. Overall, the contribution does not seem significant enough to warrant publication without strong experimental results, which this paper lacks.

---

> ### Author Response · Authors · 2020-11-19
> **Response to AnonReviewer1**
>
> Thank you for your review.
> Following your advice, we made a second experiment as in the BCQ paper. Despite HDCR being lower than methods that train policy function by taking gradient through Q-function, it provides better results than other methods that directly optimize policy function.
> About other metrics:
> Asymptotically, derivatives of both MMD and Wasserstein tend to 1. This issue can affect training, especially when learning from offline data. This problem also exists in methods that use KL. In contrast, the derivative of Hellinger distance tends to 0, providing less conservative updates.

---

> > ### Comment · AnonReviewer1 · 2020-11-23
> > **Read authors' response**
> >
> > I have read the author's response and the updated experiments. Unfortunately, the empirical results are simply not convincing at all. Even just compared against the ABM and AWR baselines, the proposed method does not demonstrate a clear advantage (final returns are often within 1-std deviation of each other, while learning curves generally look very similar with no clear winner), whlie being significantly outperformed by BCQ.
> >
> > Regarding BCQ taking the gradient through the Q-function: I don't see any inherent reason why taking the gradient through the Q function necessarily leads to better results. It could certainly lead to faster optimization, but that isn't necessarily very relevant in an offline RL setting. Additionally, in the AWR paper, their results in both online and offline experiments were reasonably competitive with RL methods that used Q-function gradients. I would recommend the authors make a more clear and explicit argument as to when using critic gradients like BCQ should perform poorly and empirically demonstrate a setting where such methods fail and methods like HDCR do.

---

### Official Review · AnonReviewer4 · 2020-10-28
**Idea is not novel enough; results are not significantly better than baselines**

**Rating:** 4
**Confidence:** 4

**Review:**

##########################################################################

Summary:


The paper provides a new metric - Hellinger distance to be combined with trust region ideas in policy optimization. The major difference from prior work is the change of this distance metric. The paper shows that with this distance metric, along with Lagrangian relaxation, one could show analytic results of improved policies. The paper also shows similar lower bound improvement results and compared with baselines on offline rl tasks.

##########################################################################

Reasons for score:


Overall, I vote for rejection. I think the idea of changing the distance metric is not novel enough. Critically, I do not think so far in the paper there is a strong enough motivation to use this distance metric: both innovation-wise and result-wise. I will explain in details below.

##########################################################################Pros:


1. Idea is not novel: the overall idea of using an alternative metric does not seem novel. Though the authors motivated an 'improved' version of the trust region lower bound, by using the fact that the Hellinger distance is upper bounded by KL - I think such an improvement in the lower bound is a bit trivial and does not provide new perspectives on the old results.

2. This new lower bound also might not provide additional benefits in practice - because in practice such lower bounds are generally too conservative.

3. Experiment results are also not strong enough. I will explain below.

##########################################################################

Cons:


1. The final performance of all three baseline algorithms are fairly bad in terms of final rewards (e.g. for halfcheetah, all returns are negative, yet we know that online algorithms could achieve >3000 at least and in some cases >6000). I wonder if this general inferior performance is a result of using offline dataset - in that sense, does the agent learn anything meaningful at all?

2. From both fig 1 and fig 2, about for half of the tasks the performance seem to drop (or stay at the same level) as the case where no training is done (x-axis at the origin). Does this also corroborate my previous concern that these agents do not learn much at all?

3. From the curves presented in Fig1,2, as well as mean+std results in Table 1,2, it does not seem that the new method provides much significant gains either.

##########################################################################

Questions during rebuttal period:

Please address and clarify the cons above. Thanks.


#########################################################################

---

> ### Author Response · Authors · 2020-11-19
> **Response to AnonReviewer4**
>
> Thank you for the review.
> 1. We do not know any papers where authors propose new metrics (except KL and Total Variation Divergence) which use is motivated theoretically and offline method for this metric is derived. The main purpose of the paper was to derive an offline method that will provide an alternative to FQI, AWR, and AWAC methods, which use KL divergence.
> 2. Actually, using Hellinger distance allow us to make bigger steps because the distance's derivative asymptotically tends to zero.

---

### Official Review · AnonReviewer2 · 2020-10-29
**Short and interesting paper, needs minor clarifications**

**Rating:** 5
**Confidence:** 2

**Review:**

This paper proposes an algorithm for off-policy reinforcement learning using the Hellinger distance between the sampling policy and optimized policy as a constraint. The motivation for the proposed method is explained in the preliminaries section. The actual algorithm and experiments run using the proposed algorithm are also provided.

The derivation is easy to follow, and this is because of the well-known lower and upper bounds on the Hellinger distance.

The writing of the paper needs work. For example, the abstract talks about the sampling policy and current policy. By current policy, what the authors mean is the policy that is being optimized. The sampling policy is the policy that was run offline. Clarifying these terms would help. Similarly, I did not follow "return for the new policy is improved comparing to KL".

In paragraph 3: "With the use of Lagrangian, have been derived" needs proofreading. In eqn 13, what is beta?

In the figures, what are the axes?

---

> ### Author Response · Authors · 2020-11-19
> **Response to AnonReviewer2**
>
> Thank you for your review.
> We carefully revised the paper and clarified the mentioned phrases as well as some others. Also, we provided a y-axis title, which is an averaged episode reward of policies during evaluation. X-axes are iterations of the training.

---

### Author Response · Authors · 2020-11-19
**Experiments update**

We would like to thank reviewers for their detailed reviews.
Reviewers noted insufficient experimental part of the paper. Therefore we reshaped the second experiment to make it more conventional. We trained a behavioral policy (DDPG) on 1M timesteps and then collected a buffer of 100k timesteps as it is in BCQ paper. All four methods used the same buffer (dataset) during the whole training.
Also, we publish individual responses below.

---

### Decision · Program_Chairs · 2021-01-07
**Final Decision**

**Decision:**

Reject

**Comment:**

The reviewer concerns generally centered around the novelty of replacing the distance metric for a policy constraint. While the authors clarified many of the reviewer concerns and added some additional comparisons, in the end it was not clear why the proposed approach was interesting: while it is true that this particular distance metric has not been evaluated in prior work, and the result would have been interesting if it resulted in some clear benefits either empirically or theoretically, in the absence of clear and unambiguous benefit, it's not clear how valuable this concept really is. After discussion, the reviewers generally found the paper to not be ready for publication in its present state.